# Silver Nanoparticles Phytofabricated through *Azadirachta indica*: Anticancer, Apoptotic, and Wound-Healing Properties

**DOI:** 10.3390/antibiotics12010121

**Published:** 2023-01-09

**Authors:** Yogesh Dutt, Ramendra Pati Pandey, Mamta Dutt, Archana Gupta, Arpana Vibhuti, V. Samuel Raj, Chung-Ming Chang, Anjali Priyadarshini

**Affiliations:** 1Department of Microbiology, SRM University, 39, Rajiv Gandhi Education City, Post Office P.S. Rai, Sonepat 131029, Haryana, India; 2Mamta Dental Clinic, Opposite Sector 29, Main Badkhal Road, Faridabad 121002, Haryana, India; 3Department of Biotechnology, SRM University, 39, Rajiv Gandhi Education City, Post Office P.S. Rai, Sonepat 131029, Haryana, India; 4Master & Ph.D. Program in Biotechnology Industry, Chang Gung University, No. 259, Wenhua 1st Rd., Guishan District, Taoyuan City 33302, Taiwan

**Keywords:** *Azadirachta indica*, phytofabrication, silver nanoparticles, anticancer drugs, caspase-3 expression, wound healing, antimicrobial agents, multidrug resistance

## Abstract

Silver nanoparticles (AgNPs) have unlocked numerous novel disciplines in nanobiotechnological protocols due to their larger surface area-to-volume ratios, which are attributed to the marked reactivity of nanosilver, and due to their extremely small size, which enables AgNPs to enter cells, interact with organelles, and yield distinct biological effects. AgNPs are capable of bypassing immune cells, staying in the system for longer periods and with a higher distribution, reaching target tissues at higher concentrations, avoiding diffusion to adjacent tissues, releasing therapeutic agents or drugs for specific stimuli to achieve a longer duration at a specific rate, and yielding desired effects. The phytofabrication of AgNPs is a cost-effective, one-step, environmentally friendly, and easy method that harnesses sustainable resources and naturally available components of plant extracts (PEs). In addition, it processes various catalytic activities for the degradation of various organic pollutants. For the phytofabrication of AgNPs, plant products can be used in a multifunctional manner as a reducing agent, a stabilizing agent, and a functionalizing agent. In addition, they can be used to curtail the requirements for any additional stabilizing agents and to help the reaction stages subside. *Azadirachta indica*, a very common and prominent medicinal plant grown throughout the Indian subcontinent, possesses free radical scavenging and other pharmaceutical properties via the regulation of proinflammatory enzymes, such as COX and TOX. It also demonstrates anticancer activities through cell-signaling pathways, modulating tumor-suppressing genes such as p53 and pTEN, transcriptional factors, angiogenesis, and apoptosis via bcl2 and bax. In addition, it possesses antibacterial activities. Phytofabricated AgNPs have been applied in the areas of drug delivery, bioimaging, biosensing, cancer treatment, cosmetics, and cell biology. Such pharmaceutical and biological activities of phytofabricated AgNPs are attributed to more than 300 phytochemicals found in *Azadirachta indica*, and are especially abundant in flavonoids, polyphenols, diterpenoids, triterpenoids, limonoids, tannins, coumarin, nimbolide, azadirachtin, azadirone, azadiradione, and gedunin. Parts of *Azadirachta indica*, including the leaves in various forms, have been used for wound healing or as a repellent. This study was aimed at examining previously biosynthesized (from *Azadirachta indica*) AgNPs for anticancer, wound-healing, and antimicrobial actions (through MTT reduction assay, scratch assay, and microbroth dilution methods, respectively). Additionally, apoptosis in cancer cells and the antibiofilm capabilities of AgNPs were examined through caspase-3 expression, dentine block, and crystal violet methods. We found that biogenic silver nanoparticles are capable of inducing cytotoxicity in HCT-116 colon carcinoma cells (IC_50_ of 744.23 µg/mL, R^2^: 0.94), but are ineffective against MCF-7 breast cancer cells (IC_50_ >> 1000 µg/mL, R^2^: 0.86). AgNPs (IC_50_ value) induced a significant increase in caspase-3 expression (a 1.5-fold increase) in HCT-116, as compared with control cells. FITC-MFI was 1936 in HCT-116-treated cells, as compared to being 4551 in cisplatin and 1297 in untreated cells. AgNPs (6.26 µg/mL and 62.5 µg/mL) induced the cellular migration (40.2% and 33.23%, respectively) of V79 Chinese hamster lung fibroblasts; however, the improvement in wound healing was not significant as it was for the controls. AgNPs (MIC of 10 µg/mL) were very effective against MDR *Enterococcus faecalis* in the planktonic mode as well as in the biofilm mode. AgNPs (10 µg/mL and 320 µg/mL) reduced the *E. faecalis* biofilm by >50% and >80%, respectively. Natural products, such as *Syzygium aromaticum* (clove) oil (MIC of 312.5 µg/mL) and eugenol (MIC of 625 µg/mL), showed significant antimicrobial effects against *A. indica*. Our findings indicate that *A. indica*-functionalized AgNPs are effective against cancer cells and can induce apoptosis in HCT-116 colon carcinoma cells; however, the anticancer properties of AgNPs can also be upgraded through active targeting (functionalized with enzymes, antibiotics, photosensitizers, or antibodies) in immunotherapy, photothermal therapy, and photodynamic therapy. Our findings also suggest that functionalized AgNPs could be pivotal in the development of a novel, non-cytotoxic, biocompatible therapeutic agent for infected chronic wounds, ulcers, and skin lesions involving MDR pathogens via their incorporation into scaffolds, composites, patches, microgels, or formulations for microneedles, dressings, bandages, gels, or other drug-delivery systems.

## 1. Introduction

### 1.1. Ulcers and Wounds

The four phases of wound healing include hemostasis, inflammation, proliferation (new tissue formation, granulation, and angiogenesis) [1,2], and tissue remodeling; these phases are capable to overlap in time and space [3]. Following the initial hemostasis, inflammation is a critical part of the normal wound-healing process [1]. Wound healing is one of the primary survival mechanisms, which is based on a very complex series of biological events or processes [1]; however, the mechanisms can be complicated if they are compromised by multidrug-resistant (MDR) microorganisms that prolong inflammation, as inflammation hampers the epithelialization and cellular process, resulting in a chronic wound (Figure 1) [4,5]. Frequently, MDR pathogens have been isolated from burns, diabetic foot ulcers, and wounds; in addition, *E. faecalis* and *S. aureus* are known for causing chronic wound infections [5,6]. Microbial biofilm is known to delay the cellular process, collagen formation, and the remodeling of ulcers and wounds [4,5,7,8]; furthermore, biofilm-led pathogens can defy macrophages in order to persist and proliferate [9]. A persistent wound infection (chronic wound) may include diverse pathogens, including *P. aeruginosa* and *S. aureus*, that have been implicated in delaying wound healing [10]; thus, the polymicrobial nature of a wound infection is widely known [11,12]. Oral infections and diseases, such as ANUG, tuberculosis, syphilis, gonorrhea, HPV, HIV/AIDS, HSV, measles, and mucormycosis, are present in ulcerative forms [13]. Neoplastic ulcers may also be found as oral manifestations of numerous oral cancers [13]. Nutritional deficiencies (scurvy, pellagra, and xerostomia) and blood disorders, such as leukemia, aplastic anemia, and cyclic neutropenia, can also manifest as ulcers. Aphthous ulcers or stomatitis are mainly associated with iron deficiency, vitamin B12 deficiency, the bacterial infection of alpha-hemolytic *Streptococcus*, and the bacterial infection of *Streptococcus sanguinis*. In addition, protein and vitamin deficiencies can affect the wound-healing process [14].

#### 1.1.1. AMR in Pathogens Causing Wounds, Oral Ulcers, and Lesions

MDR microbial species, such as *Staphylococcus* spp. and *Enterococcus* spp., have been isolated from burns, diabetic foot ulcers, and wounds [2,5,6,16]. These microbes are responsible for infections and high complications, if presented with comorbidities such as diabetes, cancer, and an immunosuppressive state [5]. Other multidrug-resistant microorganisms, isolated from infected or noninfected wounds, are *Pseudomonas aeruginosa* and beta-hemolytic *Streptococci* [2]. Delayed wound healing and wound-related infections have been attributed to microbial biofilms [1,4,17]; in addition, biofilm-led microbes are capable of countering the actions of neutrophils, macrophages, and other defense cellular processes [18]. Any disturbances or obstructions in the collagen formation and in the cellular action of fibroblasts or macrophages within the proliferation, inflammation, or remodeling stages may result in delayed wound contraction [19]. Antimicrobial resistance and metastasis of infections, to distant organs that can induce systemic infections, have also been implicated in chronic wound infections [18]. Consequently, inducing toxicity against major organs, adverse reactions, and drug antagonism can further complicate the remediation efforts [20]. After years of drug development for wound healing, the remediation strategies, for efficiently managing the skin damages, do not have an edge over MDR infections [21]. A few antimicrobials, such as silver compounds, have been examined for their bactericidal properties. Such antimicrobials have been potentially considered for treating wound infections caused by *Pseudomonas* spp., methicillin-resistant *Staphylococcus* spp., and *Enterococcus* spp. [20,22,23].

#### 1.1.2. Factors Affecting Ulcers and Wound Healing

Broadly, oxygen, bacterial infections, foreign bodies, and venous sufficiency at the local level, and diseases such as diabetes and blood disorders in addition to immunosuppressive conditions such as HIV/AIDS, radiation therapy, and cancer at the systemic level can severely affect the wound-healing process. Oxygen is essential for proper wound healing, especially for cell metabolism. Due to a high metabolism, vascular disruption, and oxygen consumption, the wound environment may become hypoxic [2,17,24,25]. Initially, the hypoxic condition may affect the secretion of cytokines, chemokines, and other regulatory proteins and factors crucial for angiogenesis, proliferation, migration, and chemotaxis [2]. However, long-term hypoxia can delay healing [17]. Chronic administration of systemic corticosteroids via the suppression of the cellular response, anti-inflammatory effects, and the reduced production of hypoxia-induced factor-1 can delay the wound-healing process. Topic corticosteroids have been shown to improve wound healing, reduce hypergranulation tissue formation, and reduce inflammation [2,26,27]. Due to antiplatelet functions, short-term, non-steroidal, anti-inflammatory drugs such as ibuprofen can delay proliferation, epithelialization, fibroblast functions, and angiogenesis, impairing wound contraction and healing [2,28].

### 1.2. Cancer and Systemic Diseases

Of the noncommunicable diseases, cancers are responsible for a higher mortality and high morbidity globally [1]. Severe toxicity against normal cells and organs, failed targeting, and AMR have presented numerous challenges in the management of cancer; in addition, recurrences and extreme after-effects are usually seen with conventional anticancer therapies. Anemia, thrombocytopenia, diarrhea, fatigue, alopecia, fertility issues, lymphedema, delirium, infection, and neutropenia are some of the common side effects associated with conventional anticancer therapies. AMR and infection by opportunistic or MDR microorganisms are major issues for patients with immunocompromised states, such as those who have AIDS, diabetes, or cancer, as they complicate the condition [29,30,31]. Malignant conditions may become critical and vulnerable, with lower 5-year survival rates [32]. Hence, most important goal of our time is the development of potent and effective antineoplastic drugs, that can also deal with the comorbidities such as diabetes-associated wounds infections. The pathogenesis of systemic infections and diseases, such as atherosclerosis, endocarditis, brachial artery endothelial dysfunction, cystic fibrosis, and systemic inflammation, has been linked to biofilm-led AMR microorganisms, where biofilm-embedded bacterial cells can show continuous, slow growth, or no growth [18,33,34,35,36].

### 1.3. Complications

Manifestations of cancer present as severe situations in theranostics, and the remediation efforts often result in unwarranted injury and inducing toxicity against vital organs. Comorbidities, confounding factors, and after-effects can further complicate anticancer therapies. Anticancer drugs are intended to target cancerous cells exclusively, but drugs failing to provide active targeting and extended retention result in severe cytotoxicity and organ failure. The roles of biofilm in disease progression are very much evident [18,37,38]. Bacteremia, the metastatic spread of an oral infection to a distant location (endocarditis, osteomyelitis, acute bacterial myocarditis, and sinusitis), and metastatic injury (myocardial infarction, chronic meningitis, and toxic shock syndrome) have been linked with circulating bacterial toxins [39]. MDR strains of *Enterococcus faecalis* bacteria have been seen in ulcers and chronic wounds; in addition, delayed wound healing and high complications have been observed due to biofilms [40,41,42]. Due to the failure of the early generation and development of antimicrobials, the quest for novel drugs has become unavoidable as efforts to control wound infection, postoperative complications, and the management of neoplastic lesions and ulcers are severely affected [43].

### 1.4. Therapeutic Strategies

Cyclosporin, tacrolimus, corticosteroids such as triamcinolone and fluocinonide, aloe vera, 1% chlorhexidine gel alone and in combination with 1% metronidazole and 2%lignocaine, and 0.2% hyaluronic acid gel are usually used as topical applications for wounds and ulcerative lesions [44]. Novel delivery systems such as biodegradable cefazolin-loaded niosomes, which are synthesized via an electrospray onto the chitosan membrane for wound-healing applications, have been found to have the ability to enhance skin regeneration by improving re-epithelialization, tissue remodeling, and angiogenesis; in addition, there was no cell cytotoxicity induced, and they were potent against biofilm-forming pathogens [45]. Apart from that, a number of nanomaterials, such as NPs, nanocomposites, scaffolds, coatings, nanocarriers, and nanogels made of metals and polymers (chitosan, polysaccharides, alginate, chitin, and polyvinyl alcohol), have been reviewed for their tissue regeneration capabilities [46]. Silver sulfadiazine (SSD) (1%), which is known for being used to treat *P. aeruginosa*-infected wounds, is more effective than PVP-I for treating wound infections [47,48]. However, SSD has also been observed delaying the wound healing and inducing toxicity in the murine fibroblast cells [27,49]. Silver-containing delivery systems, such as SSD, have also been tested extensively for wound dressings [50,51]. Poor solubility in an aqueous environment and its chemical stability have restricted the further use of SSD as a drug delivery system [15,52]. Silver in the form of nanoparticles (AgNPs) can promote wound healing by impeding biofilms and promoting the destruction of microbial membrane structures; in addition, AgNPs induce cytotoxicity against pathogens such as *Staphylococcus* spp., *Enterococcus* spp., *V. cholerae*, and *B. subtilis* [23,53]. Novel collagen and chitosan scaffolds containing AgNPs (at 10 µg/mL) were capable of healing wounds through cellular migration [54]. AgNPs were also found to be anti-inflammatory in action.

AgNPs, as well as their composites, assemblages, and complexes, are widely utilized for antimicrobial and wound-healing purposes [55]. In addition, incorporation of metals, metal oxide nanoparticles and silver-containing compounds, into gels [56,57], hydrogel or gelling fibers [23,54,58], and mesh or polymeric membranes, has been mentioned as one of the natural solutions for the development of unique bandages [58,59]. Nanogel and nanomesh, functionalized with AgNPs, growth hormones, antibiotics, or enzymes, have been suggested as wound-dressing systems, for enhancing wound healing, for decreasing the inflammatory responses, and for enhancing the immune responses [60]. AgNPs embedded in wound-dressing polymers, alginate, cotton fabrics, cellulose, or chitosan, promote wound healing and control MDR microbial growth [61,62,63]. Ag^+^ ions can create pores in the bacterial cell wall, by binding with the sulfur- and phosphorus-containing proteins of the cell wall and the cell membrane [64,65,66]. Ag^+^ ions can disrupt the cellular permeability [67,68,69], can generate oxidative stress, and can cause cell death [53,69,70,71]. The abovementioned factors are responsible for the antibacterial effects of AgNPs, and are caused by protein leakage, by compromising the cellular wall integrity, and by the inactivation of LDH through ROS production.

Surgery, chemotherapy, and radiation therapy, as the most viable and common approaches, as well as novel techniques such as immunotherapy and gene therapy, are currently considered in cancer treatment and management. The anticancer potential of NPs, including AgNPs, has already been discussed in detail in our previous study [72]. Biogenic AgNPs (14 nm), synthesized from *Podophyllum hexandrum* Royle extracts, are considered very effective against HeLa cells for inducing DNA damage and caspase-3-mediated apoptosis [73]. Functionalized AgNPs are more likely to gain intracellular access to cancer cells through a passive mode and to deliver anticancer drugs in a greater quantity via their extended permeability and retention (EPR) effects [74]. Recently, a number of novel methods and techniques capable of targeting neoplastic cells efficiently have emerged [75,76,77]. Among these unfamiliar and ingenious options, nanotechnology has provided a remarkable canvas for cancer research [77].

Natural products, such as *Azadirachta indica* (neem tree) and *Syzygium aromaticum* (clove), have been examined against *Enterococcus faecalis*, for antimicrobial effects [78,79]. In addition, they have been used in mouthwashes, toothpastes, and ointments [80,81,82] for their analgesic, anti-inflammatory, and antiseptic effects [78,83]. AgNPs are some of the most effective options available against microbial infections [84,85], due to their antimicrobial activities, their capabilities for incursion on the microbial physical structures, and their interaction with the molecular armamentariums [86,87]. AgNPs have been found to be relevant in areas such as drug delivery, bioimaging, biosensing, cancer, cosmetics, and cell biology; in addition, their applications in textiles, automobiles, air purification, and water purification are well acknowledged [88,89,90]. Nanoparticles (<100 nm) can be synthesized from inorganic (silica, quantum dots, and metal nanoparticles) or organic (liposomes, micelles, dendrimers, and polymeric nanoparticles) materials through physical, chemical, or biological approaches [91]. Being the most effective against bacterial cells and cancer cells, AgNPs have become popular for invading cells, for creating oxidative stress, and for damaging cellular structures [92]. In various forms, AgNPs have been widely applied to control wound infections [93,94,95].

### 1.5. Gaps

Gaps can be observed in the treatment of cancers with or without ulcerative lesions, since a common and potent drug is yet to be developed for cancers, chronic ulcers and wounds, and MDR microbial infections. Second, a majority of the already developed potent compounds are either cytotoxic, genotoxic, or impractical to be applied. Third, the synthesis or fabrication processes of anticancer and wound-healing compounds or drugs are either very complex or require extensive resources. Fourth, a majority of previous studies have traditionally focused on ATCC strains or strains not implicated in biofilm-led wound infection. For anticancer and improved wound-healing properties, novel drug candidates such as AgNPs have been developed by using concepts of green chemistry and a cost-effective, one-step, environmentally friendly, easy method that harnesses sustainable resources [96,97,98,99,100,101]. Biologically synthesized AgNPs show excellent hemocompatibility and antibacterial and anticancer properties as compared to chemically or polymerically synthesized NPs and commercial NPs. Therefore, a batch of assays can be performed, for analyzing the anticancer properties and for analyzing the cellular migration actions of biogenic AgNPs.

### 1.6. Aim

To examine the anticancer, apoptotic, wound-healing, and antimicrobial properties of previously biosynthesized silver nanoparticles.

### 1.7. Rationale of the Study

Due to their extremely small size, AgNPs can enter cells, interact with organelles, and yield distinct biological effects; in addition, AgNPs can contribute significantly to the design of drug delivery systems, anticancer therapies, tissue regeneration methods, and antimicrobial therapies. Therefore, we examine the anticancer, apoptotic, wound-healing, and antimicrobial properties of previously phytofabricated AgNPs.

## 2. Materials and Methods

### 2.1. Silver Nanoparticles

Nanoparticles are solid colloidal particles that are nanosized (<100 nm); due to their exemplary size and ability to confine electrons, they possess special optical and physiochemical characteristics, which are distinct from their powder, plate, or sheet forms [102,103]. AgNPs can be fabricated by using concepts of green chemistry and cost-effective, one-step, environmentally friendly, easy methods to harness sustainable resources and naturally biodegradable components, such as polysaccharides, biopolymers, vitamins, plant extracts (PEs), and microorganisms. Plant-based methods provide feasible and eco-friendly options for processing various catalytic activities that degrade various organic pollutants. Plants and plant products provide few advantages over the time-consuming and complex multistep process with a high throughput that involves yeast, fungal, or bacterial cultures. If properly fabricated and functionalized with appropriate biomolecules or drugs, NPs can bypass the immune cells, stay in the system for a longer period, provide a higher distribution, and reach the target tissue at a higher concentration. NPs can avoid being diffused to adjacent tissues and can release therapeutic agents or drugs at a specific rate on specific stimuli and for a longer duration; in addition, NPs can yield desired biological effects that can be used in imaging. The AgNPs with 44.6 nm to 66 nm diameters that were used in this study were phytofabricated previously by the authors from *Azadirachta indica* leaf extracts [72].

### 2.2. Chemicals and Reagents

Dulbecco’s modified Eagle medium that included a high-glucose (DMEM-HG) medium and an MTT (3-(4, 5-dimethylthiazolyl-2)-2, 5-diphenyltetrazolium bromide) reagent, fetal bovine serum (FBS), Hank’s balanced salt solution (HBSS), and 0.25% trypsin-EDTA solution were purchased from MP Biomedicals (Eschwege, Germany). Dulbecco’s phosphate-buffered saline (DPBS), trypsin-EDTA, a 2% paraformaldehyde solution, 0.1% Triton-X 100 in 0.5% BSA solution, 0.5% BSA in 1× PBS, hydrogen peroxide, sodium hypochlorite (3%), chlorhexidine (2%), and EDTA were procured from HiMedia (Mumbai, India). The APO-DIRECT™ kit and FITC rabbit anti-active caspase-3 IgG antibody were purchased from Pharmingen (BD Biosciences). Standard silver stock, DMSO, and Suprapur-grade 2% HNO_3_ were procured from Merck (Darmstadt, Germany). Antibiotic discs were purchased from Titan Biotech (New Delhi, India). Antibiotics and resazurin sodium salt were purchased from CDH (New Delhi, India). Clove and eugenol were purchased from Aggarwal Drug Co. (New Delhi, India). Water was obtained from the Milli-Q^®^ Integral Water Purification System (Millipore, Burlington, MA, USA).

### 2.3. Cell Lines

HCT-116, MCF-7, and V79 Chinese hamster lung fibroblast cell lines were obtained from NCCS (Pune, India).

### 2.4. MTT Cytotoxicity Assay

The cytotoxicity of AgNPs (experimental group) in HCT-116 and MCF-7 cells was analyzed using yellow tetrazolium MTT (3-(4, 5-dimethylthiazolyl-2)-2, 5-diphenyltetrazolium bromide), according to the methods explained earlier [104,105,106,107,108]. Untreated cells were considered as controls and all experiments were performed in triplicates. Cells cultured in T-25 flasks (Biolite, Thermo Fisher Scientific Inc., Waltham, MA, USA) were trypsinized, aspirated into a 5 mL centrifuge tube (Tarsons, Kolkata, India), and pelleted via centrifugation at 300× *g*. The cell count was adjusted by using the DMEM-HG medium (MP Biomedicals, Germany) so that 200 μL of suspension contained approximately 10,000 cells. To each well of the 96-well microtiter plate (Nunc, Thermo Fisher Scientific Inc., USA), 200 μL of the cell suspension was added, and the plate was incubated at 37 °C and in a 5% CO_2_ atmosphere for 24 h. After 24 h, the spent medium was aspirated and 200 μL of different test concentrations of AgNPs was added to the respective wells. The plate was then incubated at 37 °C and in a 5% CO_2_ atmosphere for 24 h. The plate was removed from the incubator and the AgNPs containing media were aspirated. Thereafter, 200 μL of the medium containing the 10% MTT reagent (MP Biomedicals, Germany) was added to each well to obtain a final concentration of 0.5 mg/mL, and the plate was incubated at 37 °C and in a 5% CO_2_ atmosphere for 3 h. The culture medium was then removed completely without disturbing the crystals formed. Subsequently, 100 μL of solubilization solution (DMSO, Merck, Germany) was added, and the plate was gently shaken in a gyratory shaker to solubilize the formed formazan. The absorbance was measured using a microplate reader at a wavelength of 570 nm and 630 nm. The percentage growth inhibition was calculated after subtracting the background and the blank. The concentration of AgNPs that was needed to inhibit cell growth by 50% (IC_50_) was calculated from the dose–response curve.

### 2.5. Caspase-3 Mode of Action

Apoptotic activities were examined (in triplicates) via the quantification of caspase-3 in AgNP-treated (IC_50_ value) HCT-116 cells, following the methods explained earlier with some modifications [109,110]. Cisplatin-treated (44 μg/mL) and -untreated cells were considered as controls. Cells were cultured in a 6-well plate (Biolite, Thermo Fisher Scientific Inc., Waltham, MA, USA), at a density of 3 × 10^5^ cells/2 mL and incubated in a CO_2_ incubator at 37 °C for 24 h. After incubation, the spent medium was aspirated and cells were washed with 1 mL 1X PBS (HiMedia, Mumbai, India). Cells were then treated with the required concentration of the experimental test compound in 2 mL of culture medium and incubated for 24 h. One of the wells was left untreated to be used as the negative control. At the end of the treatment, the medium was removed from all the wells, placed into the 5 mL centrifuge tubes, and washed with 500 μL PBS (PBS was saved in the same tubes). The PBS was removed and 200 μL of trypsin-EDTA solution (HiMedia, Mumbai, India) was added and incubate, at 37 °C for 3–4 min. The culture medium was poured back into the respective wells and cells were harvested directly into the centrifuge tubes. The tubes were centrifuged for five minutes at 300× *g* at 25 °C and the supernatant was decanted carefully. Washing with 1× PBS was performed twice. The PBS was decanted completely. Then, 0.5 mL of 2% paraformaldehyde solution was added and incubated for 20 min. Washing with 0.5% bovine serum albumin (BSA) in 1X phosphate-buffered saline (PBS) was performed. Thereafter, 0.1% Triton-X 100 in 0.5% BSA solution was added and incubated for 10 min. Washing with 0.5% bovine serum albumin (BSA) in 1X phosphate-buffered saline (PBS) was carried out 2 times. Next, 0.5% BSA in 1X PBS was added and 20 μL anti-caspase-3 antibody (BD Biosciences, Catalog No. 559341) was added and mixed thoroughly via pipetting before being incubated for 30 min in the dark at room temperature (25 °C). Once the washing with 1X PBS was performed, 0.5 mL of PBS was added, mixed thoroughly, and analyzed. If samples were not analyzed immediately, mixing was performed thoroughly just prior to analysis.

### 2.6. Scratch Assay

*In vitro* wound-healing properties of AgNPs (6.26 µg/mL and 62.5 µg/mL) and standard EGF (0.01 µg/mL) were examined (in triplicates) by using a scratch assay in accordance with the methods explained earlier with a few modifications [54,111]. V79 cells were cultured in 6-well plates to form a monolayer. As the cells reached around a 70% confluence, a scratch was made with a 200 μL pipette tip to form a wound, which was followed by the washing of the monolayer with 1 mL DPBS two times. Subsequently, 2 mL of medium was added to each well with (experimental group) or without test drugs (AgNPs), and was incubated for 24 h. Untreated wells and wells treated with 0.01 µg/mL EGF were considered as controls. AgNP-treated wells were considered as experimental. Using an inverted phase-contrast microscope, images were taken at regular time intervals (0 h, 12 h, and 24 h). The percentage of cell migration/wound healing was calculated by comparing the final gap area (24 h) to the initial gap area (0 h).

### 2.7. Microorganism

The human pathogen *Enterococcus faecalis* was isolated from the mixed culture obtained from the Centre for Drug Discovery Design and Development (C4D), Department of Microbiology, SRM University Haryana, Sonepat (India).

### 2.8. Antibiogram

The susceptibility of *Enterococcus faecalis* to 26 antibiotics from 14 different classes (including first-line, second-line, and last-line drugs) was examined (in triplicates) through the Kirby–Bauer and the microbroth dilution method [112,113]. Results were interpreted in accordance with the EUCAST and the CLSI standard procedure for antimicrobial susceptibility testing (2021) [114,115].

### 2.9. Antimicrobial Susceptibility Testing

The antimicrobial susceptibility of *Enterococcus faecalis* was investigated by utilizing the following methods in accordance with EUCAST and CLSI standard procedures [112,113,114,115]. All experiments were performed in triplicates.

#### 2.9.1. Agar Well Diffusion Method

The antimicrobial susceptibility of *Enterococcus faecalis* to AgNPs and common antimicrobials such as hydrogen peroxide (H_2_O_2_), sodium hypochlorite (NaOCl) (3%), chlorhexidine (CHX) (2%), and EDTA was examined by using the agar well diffusion method. The antimicrobial susceptibility of *Enterococcus faecalis* to clove oil and eugenol was also examined through the agar well diffusion method.

#### 2.9.2. Microbroth Dilution Method

The antimicrobial activities of AgNPs, clove oil, and eugenol against *Enterococcus faecalis* were investigated by utilizing the microbroth dilution method, which was carried out pursuant to the methods explained prior to this with some modifications [112,113]. The minimum inhibitory concentration of AgNPs opposing *Enterococcus faecalis* was established by incubating *Enterococcus faecalis* in a U-bottom, 96-well microtiter plate. AgNPs were serially diluted in sterile Mueller–Hinton broth (MHB). Aggregately, the cumulative volume attributed to each well was 100 μL as a consequence of adding 50 μL of the 24-hour bacterial culture (0.5 McFarland). The column-wise terminating concentration appertaining to AgNPs turned out to be 1280 μg/mL to 2.5 μg/mL. A total of 10 μL (0.7 mg/mL) of resazurin was added to each particular well (giving a blue or purple color) in sterile conditions and incubated for 18 to 24 h. After 6 to 8 h, the microtiter plate was examined for the transformation of color from blue to pink or red, indicating bacterial growth. *Enterococcus faecalis* ATCC 29212 was used as reference strain. The standardization of the method was based on the criteria established by the CLSI.

#### 2.9.3. Synergistic Studies via the 2D Checkerboard Method

The antimicrobial synergistic activities of AgNPs in combination with clove oil or eugenol were examined (in triplicates) through the synergistic assay using the 2D checkerboard method, which was explained earlier and was performed with some modifications [116,117]. The mixtures of AgNPs and different concentrations of clove oil or eugenol were incubated with the 24 h broth culture of *Enterococcus faecalis* (0.5 McFarland) at 37 °C for 18 to 24 h in a 96-well plate, with a final volume of 100 µl per well. Post-incubation, about 10 μL of resazurin (0.7 mg/mL) was added to each well in the same manner explained earlier and incubated for another 6 to 8 h. Subsequently, the microtiter plate was observed for a change in color. The MIC was determined as the least amount of dilution without any change in color, from blue to pink or red. Finally, the fractional inhibitory concentration (FIC) for combinations was used as an indicator to measure the effects of combined drugs. The fractional inhibitory concentration index (FIC_Index_) was determined as the inhibitory concentration of the combination divided by that of the single antimicrobial or drug, as described by Mataraci and Dosler (2012) [116]. The combination index was derived from the highest dilution of an antimicrobial combination that permitted no visible growth. The calculation method of FIC was used. *Enterococcus faecalis* ATCC 29212 was used as a reference strain.

### 2.10. Biofilm Assay

#### 2.10.1. Dentine Block Method

The antibiofilm activity (reproduced from our previous study [118]) of AgNPs against *Enterococcus faecalis* was determined (in triplicates) by utilizing the dentine block method, which was explained earlier and was performed with some modifications [119]. Human dentine blocks (5 mm × 5 mm × 1 mm) were treated for 24 h at 37 °C in groups including the control (dH_2_O), AgNPs (MIC value), and 2% chlorhexidine; additionally, blocks were treated for combinations of different concentrations of AgNPs with either clove or eugenol. After treating the dentine blocks, 50 μL of dilution from the serial dilution solutions was poured on TSA/nutrient agar plates. CFU/mL was calculated after 18–24 h of incubation at 37 °C.

#### 2.10.2. Crystal Violet Method

The antibiofilm efficacy of AgNPs was also examined via the crystal violet method explained earlier by using a 96-well microtiter plate with some modifications [120,121].

### 2.11. Statistical Analyses

Data were statistically analyzed by means of the Kruskal–Wallis rank-sum test and by employing the Microsoft Windows statistical software “R” version 4.0.2 (R Foundation), with a *p* < 0.05 being statistically significant. Results were recorded, summarized, and presented by means of descriptive statistics.

## 3. Results

### 3.1. Nanoparticles

Silver nanoparticles (44.04 to 66.50 nm), synthesized previously [72] by the authors of this study through the green method using *Azadirachta indica*, were used in this study. Phytofabricated AgNPs had a zeta potential of −55 mV. Through FTIR analyses, it was found that some potent phytochemicals, such as flavonoids and proteins from *Azadirachta indica*, had formed a strong coating or capping on the AgNPs without affecting their secondary structure and had interacted with Ag+ and NPs for the formation of AgNPs. Phytofabricated AgNPs had a strong antibacterial activity (MIC of 10 µg/mL) against the multidrug-resistant pathogen *Enterococcus faecalis*; in addition, no IC_50_ values were recorded for AgNPs and *Azadirachta indica*, signifying the negligible cytotoxicity (in V79 Chinese hamster lung fibroblast cells) of AgNPs. Near the MIC conc. of 7.812 µg/mL and 15.62 µg/mL, 91.18% and 91.44% of V79 cells were viable, respectively. At 1000 µg/mL, about 62.47% of V79 cells were viable. AgNPs did not cause any significant DNA damage in V79 fibroblast cells. The DNA degradation activity of AgNPs, analyzed through the TUNEL assay, revealed no significant increase in the overall FITC mean fluorescence intensity (MFI) and no significant increase in the DNA fragmentation index (DFI). AgNP (10 µg/mL)-treated cells had 5.45% DNA damage and a 790 MFI, as compared to 31.2% DNA damage and a 1308 MFI in cisplatin-treated cells. The apparent permeability or *P_app_* in Caco-2 cells was moderate at 5.14 × 10^−6^ cm/s, which was quantified through ICP-MS [72].

### 3.2. MTT Cytotoxicity Assay

Figure 2 shows the 24-hour treatment of AgNPs in the HCT-116 and MCF-7 cells. AgNPs were not effective against MCF-7 breast cancer cells, as no IC_50_ value was observed. However, AgNPs were effective against HCT-116, with an IC_50_ > 500 µg/mL. A significant decline in the viability of HCT-116 cells was noticed after the concentration was 250 µg/mL and cell viability was 26.17% at a concentration of 1000 µg/mL. At a concentration of 1000 µg/mL, the MCF-7 cells were 75% viable (Figure 2).

### 3.3. Caspase-3 Mode of Action

Caspase-3 expression in HCT-116 cells was quantified through flow cytometry (Figure 3) by using the FITC rabbit anti-active caspase-3 IgG antibody. A significant increase in the caspase-3 FITC-MFI was observed in HCT-116 cells treated with AgNPs (1936, 1.5-fold increase) and those treated with cisplatin (44 μg/mL) (4551, 4.5-fold increase), as compared with control (untreated) cells (1297). An increase in the percentage of active caspase-3-positive HCT-116 cells treated with AgNPs (11.4%) was observed, as compared with cisplatin (66.2%) and untreated (8.77%) HCT-116 cells. An increase in the FITC-MFI corresponds to an increase in the active caspase-3 expression and in the treated cells going for apoptosis. FITC (green) fluorescence was collected in the FL1 detector by using a 525 nm band-pass filter.

### 3.4. Scratch Assay

No significant improvement was observed (Figure 4) in the percentage of wound healing (40.2% and 33.23%) in V79 cells treated with AgNPs (6.26 µg/mL and 62.5 µg/mL, respectively), as compared with the untreated (41.95%) and EGF-treated (57.11%) cells. There was a dose-dependent reduction in the percentage of wound healing (in the form of cellular migration) in the AgNP-treated cells. This could be due to the concentration of AgNPs in the in vitro scratch assay, suggesting the need for follow-up in vivo animal experiments to reach an effective concentration for cellular migration. AgNP treatments of 6.25 µg/mL and 62.5 µg/mL decreased the migration of V79 fibroblast cells.

### 3.5. Microorganism

*Enterococcus faecalis* (n = 79) was identified from the mixed sample through standard biochemical and microbiological tests. Isolated strains were cryopreserved in glycerol stocks and kept at −20 °C/−80 °C until further use.

### 3.6. Antibiogram

The susceptibility of *Enterococcus faecalis* to antibiotics was examined and interpretated, as per the EUCAST and CLSI standard procedure for antimicrobial susceptibility testing (2021).

#### 3.6.1. Disc Diffusion Method

The susceptibility of *Enterococcus faecalis* (n = 79) to antibiotics was examined through the disc diffusion method. It was revealed that besides amoxicillin and ampicillin, *Enterococcus faecalis* was majorly resistant to streptomycin, ciprofloxacin, cotrimoxazole, sparfloxacin, cefixime, ceftriaxone, meropenem, vancomycin, tetracycline, and nitrofurantoin. Ofloxacin and norfloxacin were the most effective antibiotics against *Enterococcus faecalis*.

#### 3.6.2. Microbroth Dilution Method

Results from the disc diffusion method were validated through additional testing. The antibiotic susceptibility of *Enterococcus faecalis* to 26 antibiotics from 14 different classes, including first-, second-, and last-resort antibiotics, was tested through the microbroth dilution method. It was found that 14 antibiotics (kanamycin, novobiocin, ampicillin, amoxicillin, cefoxitin, ceftriaxone, chloramphenicol, vancomycin, erythromycin, spiramycin, metronidazole, linezolid, colistin, and polymyxin B) from 9 classes were resistant. Streptomycin, sulfamethoxazole, norfloxacin, tetracycline, and nitrofurantoin were active against and capable of inhibiting *Enterococcus faecalis* in a planktonic state after 18–24 h of treatment. High-level antibiotic resistance (HLAR) was observed for amoxicillin (>1024 µg/mL) and novobiocin (512 µg/mL). *Enterococcus faecalis* was very sensitive to fluroquinolones, as a very small concentration (0.25 µg/mL) was able to inhibit *E. faecalis* completely. The second most effective drugs (1 µg/mL) were minocycline and azithromycin.

### 3.7. Antimicrobial Activities

#### 3.7.1. Agar Well Diffusion Method

AgNPs (ZOI (mm), 24.39 ± 4.03) were more effective (Kruskal–Wallis rank-sum test statistics: 103.06, df:2, *p* < 0.05), against *Enterococcus faecalis* (n = 79), than clove oil (21.49 ± 3.63) or eugenol (16.95 ± 2.89). Whereas, H_2_O_2_ exhibited the highest ZOI (32.68 ± 5.44) as compared to NaOCl (14.94 ± 7.06), CHX (15.94 ± 2.87), and EDTA (16.50 ± 2.9).

#### 3.7.2. Microbroth Dilution Method

The antimicrobial susceptibility of *Enterococcus faecalis* (n = 79) to AgNPs, clove oil, and eugenol was examined through the microbroth dilution method. AgNPs had an MIC of 10 μg/mL. The MICs of clove oil and eugenol were 312.5 µg/mL and 625 µg/mL, respectively. There was a statistically significant difference between the MICs of clove oil, eugenol, and AgNPs (Kruskal–Wallis rank-sum test statistics: 184.49, df:2, *p* > 0.05).

#### 3.7.3. Synergistic Studies via the 2D Checkerboard Method

The synergistic effects of AgNPs when combined with clove oil or eugenol were evaluated by determining the fractional inhibitory concentration (FIC) index. The synergistic and additive effects of AgNPs with clove oil or eugenol were observed against multidrug-resistant *Enterococcus faecalis*. However, no antagonistic effect was observed. There was a 4- to 8-fold reduction in the AgNP MIC when combined with clove oil. In addition, there was a 4- to 16-fold decline in the AgNP MIC when combined with eugenol.

### 3.8. Biofilm Assay

The antimicrobial effects of AgNPs against *Enterococcus faecalis*-formed biofilms were studied via the CFU method and crystal violet method by using human dentine blocks and a 96-well plate, respectively.

#### 3.8.1. Dentine Block Method

A significant difference (Kruskal–Wallis rank-sum test statistics: 73.278, n:99, df:8, *p* < 0.05) was observed, with a mean colony count of 9.9, 14.9, and 9.4 CFU/mL (10^7^), for the 20 μg/mL AgNPs, 10 μg/mL AgNPs, and 2% CHX, respectively. The biosynthesized AgNPs were as effective as the 2% CHX against *Enterococcus faecalis*.

#### 3.8.2. Crystal Violet Method

The antibiofilm activities of biosynthesized AgNPs were also analyzed through the crystal violet method. *E. faecalis* isolates (n:79) were strong (n:11), moderate (n:24), weak (n:35), and non-biofilm formers (n:9). Strong biofilm formers were considered for the crystal violet antibiofilm assay. Effective and dose-dependent antibacterial activities (biofilm reduction >50% and >80%) of AgNPs (10µg/mL and 320µg/mL, respectively) were observed against MDR *Enterococcus faecalis* in the biofilm mode (Kruskal–Wallis rank-sum test statistics: 107.15, df:9, *p* < 0.05).

## 4. Discussion

This study signifies that phytofabricated silver nanoparticles are effective against cancer cells and MDR human pathogens; in addition, they are capable of inducing apoptosis in cancer cells. Although the wound-healing actions were not significant, AgNPs induced weak cellular migration. Cellular migration can be improved with the appropriate functionalization, capping, or conjugation of AgNPs with other biomolecules. Prototypical features of phytofabricated nanostructures, including NPs, would provide an advantageous edge over conventional methods in the therapy of carcinomas and tumors. Such leverages are due to the ability of NPs to reach target cells or tissue without diffusing to the adjacent areas. Such typical features are not enjoyed with conventional diffusing or anticancer therapeutic agents; furthermore, conventional therapeutic agents usually precipitate unwanted after-effects and induce cytotoxicity against normal healthy cells. Conventional drugs or therapeutic agents can target both cancerous and healthy cells. However, phytofabricated-NPs are formulated to only reach cells of interest. The findings of this study might also help in understanding cellular pathways, signaling, and disease progression through the efficient identification of novel biomarkers and mechanisms of drug action. Observations from this study also provide a scope for the modification of NPs by conjugating bioactive molecules such as enzymes, photosensitizers, therapeutic drugs, and even nucleic acids. We believe that our findings show the untapped potential of phytofabricated NPs in the areas of cancer diagnosis and prevention, antimicrobial therapies, and the prevention of morbidities.

This study was a continuation of our previous studies [72,118]. We biologically synthesized AgNPs and concluded that they had excellent biocompatibility with negligible cytotoxicity (IC_50_ >> 1000 µg/mL) in noncancerous V79 Chinese hamster lung fibroblast cells; in addition, there was no significant genotoxicity in noncancerous V79 cells due to the biosynthesized AgNPs. In the current study, the cytotoxicity of AgNPs capped with *Azadirachta indica* was examined for two cancer cell lines (HCT-116 and MCF-7) through an MTT assay (Figure 2). Our results were in line with a study by Guilger-Casagrande et al. (2019), in which no IC_50_ values were recorded for AgNPs synthesized through *Trichoderma harzianum* and that were examined through a tetrazolium reduction assay (MTT) on three cell lines [122]. Likewise, Składanowski et al. (2016) revealed the biologically synthesized AgNPs in L929 mouse fibroblasts at a concentration of 10 µg/mL to not have no cytotoxicity [123]. In our study, the IC_50_ of AgNPs against HCT-116 colon cancer cells (Figure 2) was 744.23 μg/mL, despite any significant actions against MCF-7. The significant cytotoxicity of biogenically synthesized AgNPs from *Calligonum comosum* and *Azadirachta indica* was observed on HepG2 hepatocellular carcinoma, LoVo colon adenocarcinoma, and MDA-MB231 human breast adenocarcinoma cells [124]. The IC_50_ of *A. indica*-fabricated AgNPs against HepG2 cells was 16.4 µg/mL. In addition, the IC_50_ of AgNPs synthesized from aqueous extracts of *A. indica* was recorded at 10.9 µg/mL against LoVo and MDA-MB231 cells [124].

Similarly, biologically synthesized AgNPs (from methanolic extracts of *A. indica* barks) at a very low concentration (IC_50_ of 8.02 µg/mL) were as effective as doxorubicin (IC_50_ of 6.37 µg/mL) against DU-145 human prostate cancer cells [125], suggesting other parts of medicinal plants should also be examined. *A. indica*-functionalized AgNPs have been observed inhibiting the HT1080 fibrosarcoma cancerous cells effectively, but not the HEL293 human embryonic kidney noncancerous cells [22]. This suggests that biogenic NPs are biosafe despite their toxicity to cancerous cells [126]. A similar trend was observed in our studies, where biogenic AgNPs were cytotoxic against HCT-116 cancerous cells, but not against V79 noncancerous cells [72]. AgNPs, biosynthesized using *Juglans regia* aqueous extracts, have been observed inhibiting 70% of the MCF-7 cancerous cells at 60 µg/mL, as compared to a 15% inhibition of the L929 noncancerous cells; moreover, the same AgNPs at 10 µg/mL inhibited 50% of the MCF-7 cells, but did not induce any significant cytotoxicity against noncancerous cells at that concentration [127]. It seems that the sensitivity of cancer cells to biogenic AgNPs is higher than that of other cells, such as noncancerous cells [127,128]. This was also observed with AgNPs biosynthesized with *Cordia myxa* leaf extracts [129]. The inhibition of HCT-116 and SW480 colon cancer cells was >70% at a concentration of 50 µg/mL, which was significantly higher than that in our findings (IC_50_ of 744.23 µg/mL) [129].

The selective cytotoxicity of biogenic AgNPs against cancerous cells has been suggested due to the abnormal metabolism, high proliferation rate, and higher uptake of AgNPs by cells [130]. Such factors make cancerous cells more vulnerable [130]. Functionalization also supports the unique actions of biogenic AgNPs [131]. In addition, the material utilized, the fabrication approach [127,128], the parameters of the bioassay [127], the parameters and physical characteristics of NPs [132], functionalization, and cellular conditions can usually affect the biological actions of NPs [127,128,129]. However, biogenic AgNPs can act against cancerous cells in a number of ways (Figure 5). AgNPs are more cytotoxic to cancerous cells than to noncancerous cells, comparatively [127,128,133]. We must verify our findings with additional in vivo animal experiments. Second, despite the observations of the observed cytotoxicity against particular cancer cells, an examination of in vitro cytotoxicity should be conducted on other cancer cells before reaching any conclusions. Third, AgNPs must be functionalized to enhance the anticancer effects.

Biogenic AgNPs (IC_50_ value) were also examined for their capability to induce apoptosis in HCT-116 cancerous cells. The apoptotic pathway is more likely to be activated by AgNPs, anticancer drugs, or radiation. The caspase-3 activities in HCT-116 cells were 1.5-fold higher, comparatively. Caspase-3 expression in HCT-116 cancerous cells were in coherence with the previous studies [134,135,136]. Similarly, there was a 1.18-fold increase in the caspase-3 expression in MCF-7 cells treated with AgNPs (10 µg/mL) and synthesized with *Rubus fairholmianus* extract; in addition, AgNPs induced apoptosis through the intrinsic pathway [136]. Biogenic AgNPs (12–41 nm) synthesized with *Solanum trilobatum* extracts upregulated the caspase-3 expression in MCF-7 cancerous cells [137]. Apoptotic pathway can also be initiated, by UV rays, gamma rays, or by oxidative stress from reactive oxygen species, for releasing cytochrome-c from the mitochondria and for activating the caspase-9 (Figure 6) [136,137,138]. Biogenic NPs have been considered to be very potent against carcinoma cells. AgNPs (53 nm) synthesized with *Beta vulgaris* L root extracts induced caspase-3 activities (5 µg/mL, 20 µg/mL, and 40 µg/mL) in HuH-7 human hepatic cancerous cells at a level much higher than in CHANG normal human hepatic cells [139]. In the same study, more chromosomal condensation was also observed by using DAPI, in 40 µg/mL AgNPs exposed HuH-7 cells [139]. High toxicity was detected in cancerous cells by FACS, on exposure to higher concentration of biogenic AgNPs (40 µg/mL) [139]. The biological actions of AgNPs (7–20 nm) have been examined for the induction of apoptotic activities in cancerous cell lines [140]; AgNPs were able to induce apoptosis at amounts as low as 0.78 µg/mL and 1.56 µg/mL (for HT-1080 and A431, respectively). Biogenic AgNPs (73.37 nm, 12.35 μg/mL) synthesized from *Fagonia indica* were able to induce caspase-3 in human breast cancer cells [141]. The abovementioned studies indicate that biogenic AgNPs can induce apoptosis in cancerous cells by activating caspase-3 more efficiently than in noncancerous cells.

Figure 4 shows that AgNPs were able to close the wound by migrating toward the wound space. There was no significant percentage of wound healing when comparing the lowest dose of AgNPs (40.2% for 6.26 µg/mL) with the controls. Wound healing was lowest (33.23%) for the highest dose of AgNPs (62.5 µg/mL). Capped biomolecules from *A. indica* must have affected the cellular migration and biological movement of the fibroblasts. Previous studies have examined a number of drug delivery systems by employing silver in various forms, such as NPs, in order to achieve higher wound-healing actions both in vivo and in vitro [142,143,144]. Composites of AgNPs amalgamated in polymers have been seen to have increased cellular migration and a slowed down immune response; in addition, composites or scaffolds have been seen to have enhanced epithelialization and remodeling [142]. The most accepted and possible theory for the higher wound-healing actions of AgNPs can be explained by the evidence of enhanced cellular migration, greater wound closure, cellular upregulation, and reduced toxicity against normal cells [144]. Formulations and amalgamations of silver, through different drug delivery systems, have been examined for ulcerative lesions [145,146], charred lesions [147,148], and lacerations [149]. Such formulations have been found to be very effective for assisting in faster healing, for providing analgesic effects, and for the ease of application. However, few of these formulations have been connected with cell death, skin damage, and the development of resistance in wound microbes [147,150]. Long-term exposure to silver-based formulations may result in genotoxicity, damage to kidneys, and damage to other vital organs [151].

Among all elements, silver is considered to be one of the dynamic elements due to its applications in semiconductors, nanosensors, and therapeutic solutions for ulcers and wounds [152,153,154]. Silver has been employed, through diverse forms, in different formulations and drug delivery systems. Silver sulfadiazine is one of the widely applied formulations for a number of conditions, such as chronic wounds and ulcers [142,155]. Although we found silver (in the form of nanoparticles) to be comparatively biosafe [72], its application has drawn a few concerns due to its dose-dependent and cell-dependent biological actions [108,156,157]. Silver has been observed to induce a considerable loss of viability in two noncancerous cell lines [157]. In addition, significant toxicity was induced by polymer-functionalized AgNPs in noncancerous IMR-90 and U251 cell lines [108]. AgNPs (~250 nm) have been found to induce renal toxicity in Wistar rats upon oral exposure [158]. Very significant degenerative changes were noticed in 30 mg/kg and 125 mg/kg treatment groups, upon the histopathological examination of glomeruli, such as loss of tubular architecture, loss of brush border, and interrupted tubular basal laminae [158]. On the other hand, apoptotic activities were very evident in the groups treated with 125 mg/kg and 300 mg/kg. The expression of TNF-α mRNA and EGF mRNA was also noticed in AgNP treatment groups [158]. NPs have been seen to have a dose-dependent cytotoxicity against noncancerous cells [22,159]. AgNPs biosynthesized with a *Streptomyces sp. NH28* biomass exhibited low viability (82.9 ± 7.5 %) in mammalian cells at 25 µg/mL (IC_50_ of 64.5 μg/mL) [123]. Starch-stabilized AgNPs (20 nm) induced a decline in the viability of murine cells at 10 μM [105]. Starch-capped AgNPs have been found to induce genotoxicity in IMR-90 human lung fibroblasts cells, although the cells were unaffected beyond 100 μg/mL [108]. A significant toxicity in murine hepatocytes was observed for commercial AgNPs (15 nm and 100 nm, 5 to 50 µg/mL), as compared to NPs of manganese oxide, molybdenum, aluminum, iron oxide, or tungsten [160]. A dose-dependent inhibition was also observed in RAW264.7 macrophages, with a significant cytotoxicity and significant changes in the cellular morphology being due to Cs-AgNPs [67]. It is very much evident that biological actions and behaviors of NPs are established through an array of factors, such as their functionalization, materials used in fabrication, physical parameters, or drug delivery methods [156,161].

Formulations and composites of silver have been scrutinized for their actions, such as the rate of cellular migration. Silver formulations have also been found to have poor wound-healing properties, slowed down cellular proliferation, and slowed down cellular migration [156]. However, such findings of deferred cellular movement and remodeling can be relative. Such drawbacks and hinderances can be dealt with via composites, scaffolds (illustrated in Figure 7) [144,145], nanoparticles, and nanofibers that contain silver nanoparticles [162].

Intrinsic resistance in *Enterococcus faecalis*, for beta-lactams, for clindamycin and for aminoglycosides, is well known [163,164]. In our study, resistance in *Enterococcus faecalis*, for beta-lactams, was nonconflicting with the previous studies [165,166,167]. Ofloxacin, followed by cefixime, came to be the most efficient antibiotic opposing *Enterococcus faecalis*. The susceptibility of ofloxacin was comparable with the global susceptibility rate of 44.8% [165]. No other antibiotics were as efficient as ofloxacin. However, the inefficiency of tetracycline came to be more substantial globally, relative to the prior range of 13.8% to 65.3% [164,167,168]. Norfloxacin turned out to be the second least effective antibiotic, following after the majority of the beta-lactam antibiotics. The susceptibility rate of norfloxacin was not in accordance with the global rate of 16.6% to 73.2% [169,170].

The antimicrobial effects of clove oil, eugenol, and AgNPs were evaluated for synergistic activities, by 2D checkerboard method. The combination of AgNPs, with clove or eugenol, led to a reduction in the MIC of AgNPs. AgNPs exerted an 8- and 16-fold increase within the antibacterial activities when combined with clove oil or eugenol. Silver nanoparticles had the better antimicrobial activity toward the tested MDR pathogen, but once examined at its MIC, it was comparable to clove oil and eugenol. Clove oil and its component eugenol exhibit antibacterial activities, opposing a variety of pathogens [78,79,171]. This current in vitro study demonstrated the higher inhibitory effects of clove oil as compared to eugenol against multidrug-resistant *Enterococcus faecalis*, which were greater than those recorded earlier [171,172,173]. In addition, a statistically significant difference was observed (*p* < 0.05) for the antimicrobial effects of AgNPs (MIC of 10 µg/mL), clove oil, and eugenol. Our attempts have resulted in preliminary, yet pivotal, outcomes, providing a map for replication. AgNPs have shown excellent antibacterial activity [174] against multidrug-resistant bacteria, and it can be concluded that they can be an ideal approach for treating multidrug-resistant *Enterococcus faecalis.* AgNPs in a suspension form have been observed to be more effective against *Enterococcus faecalis* as compared to a gel form [175]. Nanosilver gel (0.02%) was more effective than the nanocadmium gel (0.02%) against *Enterococcus faecalis* [176]. Halkai et al. (2018) demonstrated strong antibiofilm effects of silver nanoparticles (30 µg/mL) against *Enterococcus faecalis* [119]. Their toxicity to human tissue has also been studied. For 80–100 nm silver nanoparticles, no apparent cytotoxicity has been observed against fibroblasts [177].

Aviv (2016) reported an MIC of clove oil of 12.5 mg/mL against *Enterococcus faecalis*; this result indicates that our concentration was much more effective [178]. Additionally, Krishnan et al. (2015) reported an MIC of AgNPs of 5 mg/mL against *Enterococcus faecalis* [179], which is still less effective than our findings. Conversely, Charannya et al. (2018) reported an MIC of AgNPs at 0.015 mg/mL. These results indicate possible variations in the particle size or synthesis methods of AgNPs [180]. Synergistic as well as additive effects for AgNP–clove oil and AgNP–eugenol combinations suggest strong evidence for the combination of diverse novel agents against multidrug-resistant *Enterococcus faecalis*. It can be noted that no antagonistic effect was observed. Similar results were reported by Elshinawy et al. (2018) for AgNPs in combination with ozonated olive oil against *Enterococcus faecalis* [181].

Biosynthesized AgNPs (10 μg/mL) are as effective as chlorhexidine against *Enterococcus faecalis*, with mean CFU/mL (10^7^) values of 14.9 and 9.4, respectively; in addition, AgNPs were able to significantly reduce the formation of microbial biofilms grown in a 96-well plate at the MIC level. These results were in accordance with those of AgNPs that were biologically synthesized with the endophytic fungi *Fusarium semitectum* and isolated from healthy leaves of *Withania somnifera* (ashwagandha or winter cherry) [119]. AgNPs (30 μg/mL) have been shown to exhibit effective antibacterial activity against *Enterococcus faecalis* (ATCC 29212) biofilms [119]. Similar results were observed for commercial AgNPs (20 nm) used as a vehicle for calcium hydroxide and as 0.2% and 0.1% gels for a 7-day treatment against *Enterococcus faecalis* biofilms [175,182] The 0.2% and 0.1% AgNP gels were found to be more effective with a 1-day treatment against *Enterococcus faecalis* biofilms [183]. The results of the current study indicate the effective antibacterial activity of AgNPs (10 μg/mL) against *Enterococcus faecalis* biofilms. AgNPs that were biologically synthesized using *Bacillus licheniformis* have been shown to have significant antibiofilm effects against *Pseudomonas aeruginosa*-formed and *Staphylococcus epidermidis*-formed biofilms [121].

## 5. Conclusions

The present study shows that biosynthesized silver nanoparticles induced concentration-dependent cytotoxicity and caspase-3 apoptotic cell death in HCT-116 human colon cancer cells. AgNPs induced the significant expression of the caspase-3 FITC-MFI in HCT-116 cells, which was not induced in cisplatin-treated and untreated control cells. An increase in the FITC-MFI active caspase-3 expression in treated cells, corresponded to the induction of apoptosis in cells. The scratch assay revealed no significant improvements in wound healing due to AgNPs, as was observed in EGF-treated and untreated cells. However, the migration of V79 Chinese hamster lung fibroblasts was noticed. AgNPs possess significantly higher antibacterial activities against the MDR pathogen *Enterococcus faecalis* when it is in both a planktonic state and antibiofilm state. Clove oil and eugenol were able to inhibit *Enterococcus faecalis*. AgNPs had a significantly higher ZOI as compared to clove oil and eugenol. Combination studies (with clove oil) showed synergistic effects, with a 4- to 8-fold reduction in the MIC of AgNPs. A significant reduction in biofilms grown on dentine blocks, was observed for AgNPs, which were more potent than the known, effective antibiofilm agent 2% chlorhexidine. The reduction (>50%) in the formation of biofilms grown in 96-well plates was observed through the crystal violet method with the treatment of AgNPs (10 µg/mL). AgNPs at 320 µg/mL inhibited >80% of biofilm formation.

## Figures and Tables

**Figure 1 antibiotics-12-00121-f001:**
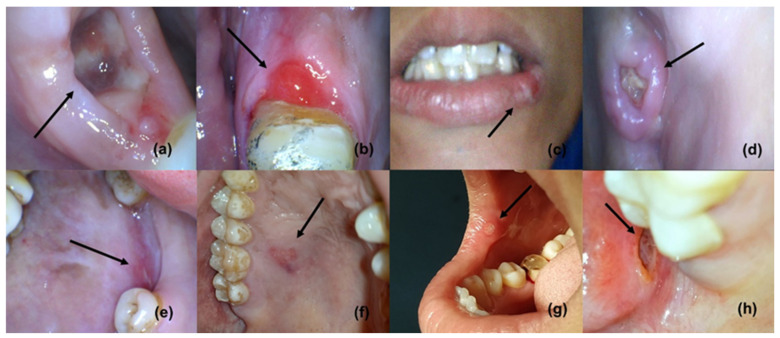
Oral ulcers and wounds (reproduced with permission from all authors) [15].

**Figure 2 antibiotics-12-00121-f002:**
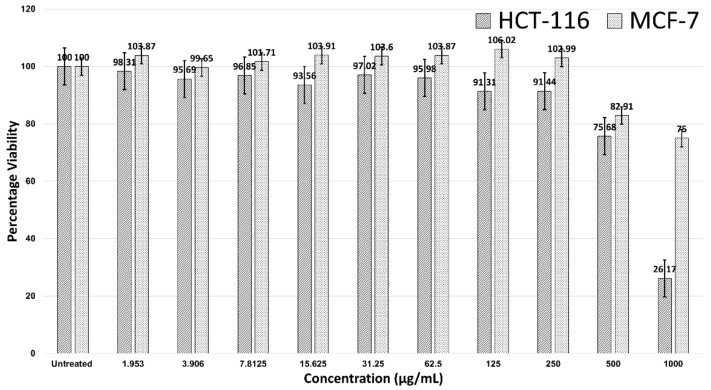
Cell viability of HCT-116 and MCF-7 cells in treatment with AgNPs.

**Figure 3 antibiotics-12-00121-f003:**
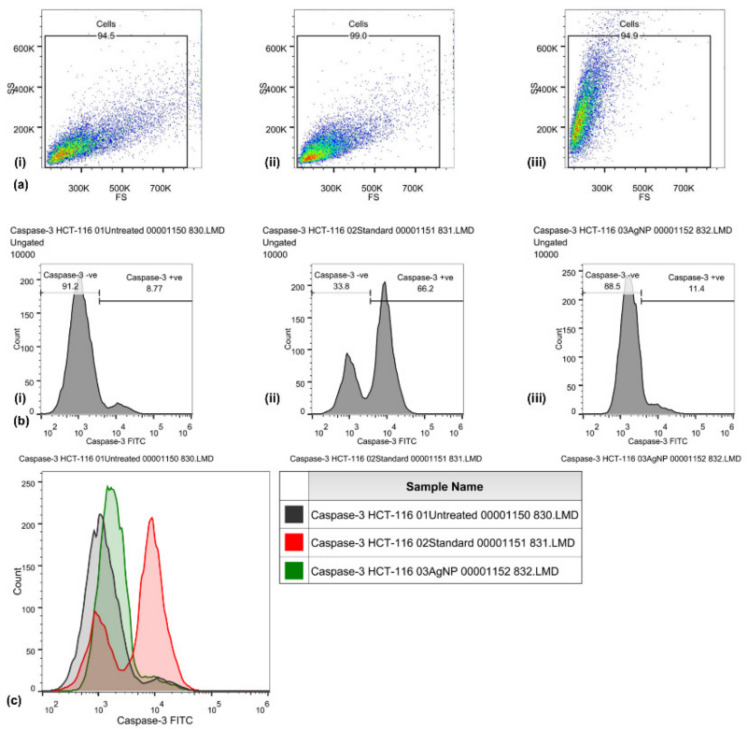
Caspase-3 activities induced by biosynthesized silver nanoparticles in HCT-116 cells, quantified through FITC rabbit anti-active caspase-3 IgG antibody (BD Biosciences). Representative FS/SS dot plots (**a**) of uniform population, percentage of cells with high caspase-3 activity assessed by dUTP-FITC (**b**) for control (i), standard EGF (ii), and biosynthesized silver nanoparticles (AgNPs) (iii). Flow cytometry histogram overlays (**c**) show the intensity of dUTP-FITC in the HCT-116 cells that are untreated (control), those treated with cisplatin (standard), and those treated with biosynthesized silver nanoparticles (AgNPs).

**Figure 4 antibiotics-12-00121-f004:**
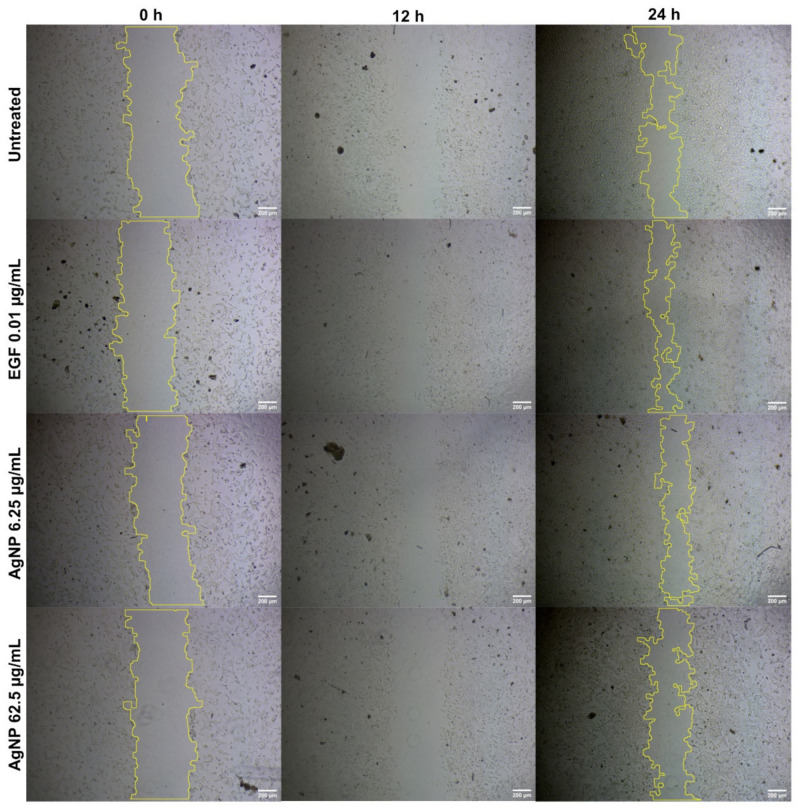
Cellular migration in V79 Chinese hamster lung fibroblast cells analyzed through scratch assay for wound-healing properties of AgNPs at different concentrations.

**Figure 5 antibiotics-12-00121-f005:**
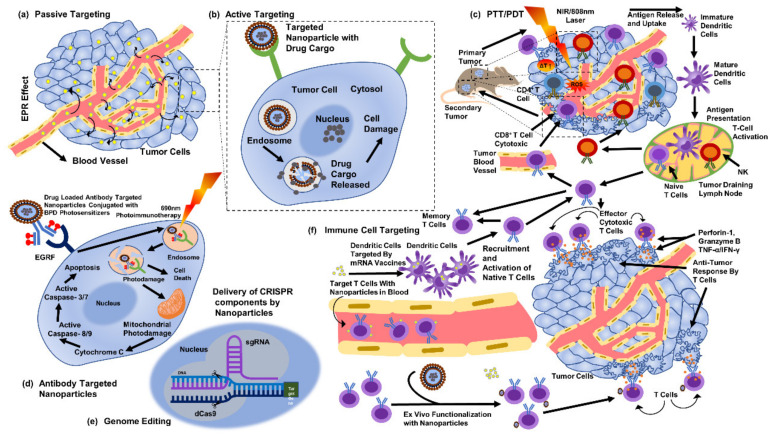
Methods to target cancer cells through silver nanoparticles (part of the figure reproduced with permission from all authors) [15].

**Figure 6 antibiotics-12-00121-f006:**
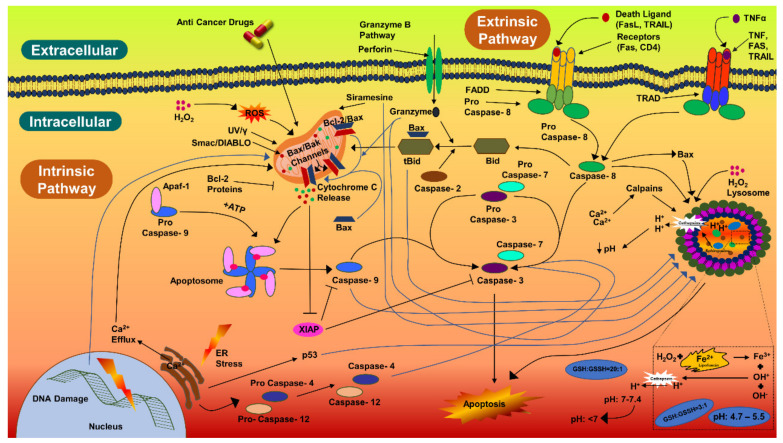
Caspase-3 pathway of cellular apoptosis.

**Figure 7 antibiotics-12-00121-f007:**
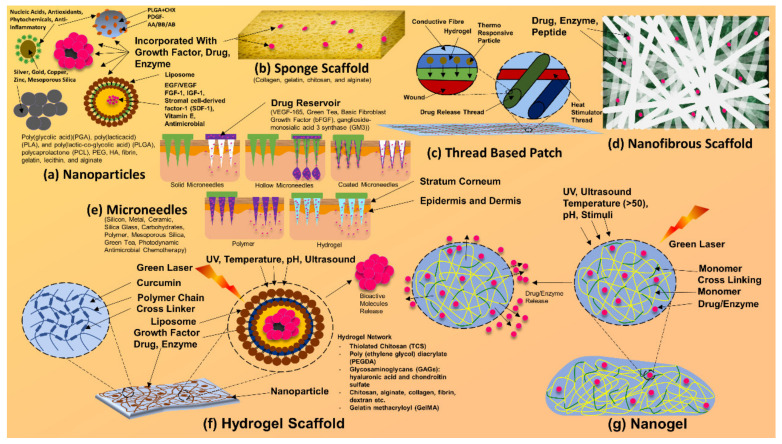
Composites and scaffolds of silver nanoparticles. Silver nanoparticle-based bandages, dressings, patches, and other drug delivery systems.

## Data Availability

Data will be made available upon request.

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
