# Peer review of "Silver Nanoparticles Phytofabricated through Azadirachta indica: Anticancer, Apoptotic, and Wound-Healing Properties"

_antibiotics, 2023, doi:10.3390/antibiotics12010121_

Round 1
Reviewer 1 Report
The MS entitled “Silver nanoparticles phyto-fabricated through Azadirachta indica: Anti-Cancer, Apoptotic, and Wound Healing Properties” was thoroughly reviewed. The article is acceptable however, the following suggestions/corrections must be addressed by the authors before further decision
1. The abstract should be started with the initial sentences that reflect the medicinal properties of Azadirachta indica from the literature.
2. The abstract should be ended with conclusive remarks based on the results that reflects future perspective.
3. Why the manuscript introduction is written in topics like observed in review articles.
4. The author should provide the rational for conducting this study in the last para of introduction.
5. A lot of work has been published on the synthesis of Ag NPs utilizing Azadirachta indica. What is the novelty of your current work?
6. IC50 should be written as IC50 throughout the manuscript.
7. Section 2.1: the author should briefly explain the significances of nanoparticle in a short para.
8. Cite the following articles to improve the scientific quality of the article;
https://doi.org/10.1590/1519-6984.257622
https://doi.org/10.1016/j.arabjc.2017.05.011
https://doi.org/10.29356/jmcs.v65i3.1479.
9. Section 4.14.3: Synergistic Studies 2D Checkerboard Method. Font size is different from rest of the test in other sections. It should be adjusted uniformly.
10. Methods in general lack references and should be supported by appropriate references.
11. Figure 2: the resolution of the figure is too low to be read the legends and values properly.
12. The resolution of figure 3 should be enhance for better readability.
13. The discussion section should be initiated with para showing significances of the study.
14. In methods: Most of the methods lack references. Here each section should be supported by suitable appropriate reference to make it more authentic.
15. Conclusion of the study is too lengthy. It should be concise on the bases of key findings.
16. The article should be revised thoroughly for spellings and grammatical mistakes.
Author Response
- The abstract should be started with the initial sentences that reflect the medicinal properties of Azadirachta indicafrom the literature.
Response: It has been corrected.
- The abstract should be ended with conclusive remarks based on the results that reflects future perspective.
Response: It has been corrected.
- Why the manuscript introduction is written in topics like observed in review articles.
Response: We have covered background and previous work in the area.
- The author should provide the rational for conducting this study in the last para of introduction.
Response: We have provided the rational for conducting this study.
- A lot of work has been published on the synthesis of Ag NPs utilizing Azadirachta indica. What is the novelty of your current work?
Response: Although numerous publications have reported about the synthesis of AgNPs utilizing Azadirachta indica, but we have reported biological as well as physical characteristics extensively of the biocompatible and novel AgNPs. Previous publications not have been reported this in details as per our knowledge.
- IC50 should be written as IC50 throughout the manuscript.
Response: It has been corrected.
- Section 2.1: the author should briefly explain the significances of nanoparticle in a short para.
Response: We have added the significance of nanoparticles in section 2.1 briefly.
- Cite the following articles to improve the scientific quality of the article; https://doi.org/10.1590/1519-6984.257622
https://doi.org/10.1016/j.arabjc.2017.05.011
https://doi.org/10.29356/jmcs.v65i3.1479.
Response: we have cited these articles too.
- Section 4.14.3: Synergistic Studies 2D Checkerboard Method. Font size is different from rest of the test in other sections. It should be adjusted uniformly.
Response: It has been corrected.
- Methods in general lack references and should be supported by appropriate references.
Response: It has been corrected.
- Figure 2: the resolution of the figure is too low to be read the legends and values properly.
Response: It has been replaced with the higher-resolution figure.
- The resolution of figure 3 should be enhance for better readability.
Response: It has been replaced with the higher-resolution figure.
- The discussion section should be initiated with para showing significances of the study.
Response: It has been corrected as suggested.
- In methods: Most of the methods lack references. Here each section should be supported by suitable appropriate reference to make it more authentic.
Response: It has been corrected.
- Conclusion of the study is too lengthy. It should be concise on the bases of key findings.
Response: It has been corrected.
- The article should be revised thoroughly for spellings and grammatical mistakes.
Response: It has been corrected.

Reviewer 2 Report
In this study, the authors investigated the anti-cancer and antimicrobial properties of phyto-fabricated silver nanoparticles, as well as their potential for wound healing applications. Revisions are required. The manuscript should be revised in response to the comments.
1. The "Abstract" would be more coherent if the authors began with silver nanoparticles, their physicochemical properties, the advantages of phyto-fabrication for producing these nanoparticles, and the role of biosynthesized silver nanoparticles in biomedical applications. The aim of the study, as well as the methods and results, should then be included. A clear concluding statement is also required at the end.
2. "Azadirachta indica" and "phyto-fabrication" are appropriate keywords for this study.
3. In the "Introduction" section, a background on the wound healing process, chronic wounds, and the role of silver nanoparticles in the treatment of hard-to-heal wounds should be provided.
4. On page 3, lines 124-127. Other common strategies, such as delivery systems (for the reference: https://onlinelibrary.wiley.com/doi/full/10.1002/jbm.b.35039) and the use of nanobiomaterials in the treatment of chronic wounds (for the reference: https://www.mdpi.com/1996-1944/12/13/2176), should be mentioned as well. It would be more in line with the study's objectives.
5. On page 5, subsection “2.1. Silver nanoparticles”, it would be helpful to provide a brief explanation of the silver nanoparticles biosynthesis method.
6. Before explaining the MTT assay, the cell culture procedure should be thoroughly explained. The MTT assay should be explained as well.
7. The number of replicates for each test should be specified.
8. Reference 51 is not available because the article is "in press." Therefore, the quality of the produced silver nanoparticles cannot be evaluated based on the current manuscript. It is strongly advised that some of the results concerning the characterization of the prepared nanoparticles be included.
9. Figure 3 is of poor quality.
10. Statistically significant changes should be indicated in figure 2.
11. The control and experimental groups of the MTT assay and scratch assay should be defined in the "Materials and Methods" section.
12. It would be interesting if the authors discussed the limitations of using silver nanoparticles and their dose-dependent toxicity in the "Discussion" section. (For the reference: https://bmcnephrol.biomedcentral.com/articles/10.1186/s12882-021-02428-5)
13. Most of the references are not up-to-date. The authors should use the most recently published articles to provide background on the studied subjects and to discuss the findings.
14. "et al." should be written in italics throughout the manuscript.
Author Response
In this study, the authors investigated the anti-cancer and antimicrobial properties of phyto-
fabricated silver nanoparticles, as well as their potential for wound healing applications.
Revisions are required. The manuscript should be revised in response to the comments.
- The “Abstract” would be more coherent if the authors began with silver nanoparticles, their physicochemical properties, the advantages of phyto-fabrication for producing these nanoparticles, and the role of biosynthesized silver nanoparticles in biomedical applications. The aim of the study, as well as the methods and results, should then be included. A clear concluding statement is also required at the end.
Response: It has been corrected as suggested.
- Azadirachta indica” and & “phyto-fabrication” are appropriate keywords for this study.
Response: It has been corrected as suggested.
- In the “Introduction” section, a background on the wound healing process, chronic wounds, and the role of silver nanoparticles in the treatment of hard-to-heal wounds should be provided.
Response: It has been corrected as suggested. Wound healing process in section 1.1, chronic wound in section 1.1.2, and roles of silver nanoparticles in hard-to-heal wounds have been included in section 1.4 of the introduction.
- On page 3, lines 124-127. Other common strategies, such as delivery systems (for the reference: https://onlinelibrary.wiley.com/doi/full/10.1002/jbm.b.35039) and the use of nanobiomaterials in the treatment of chronic wounds (for the reference: https://www.mdpi.com/1996-1944/12/13/2176), should be mentioned as well. It would be more in line with the study’s objectives.
Response: It has been included as suggested.
- On page 5, subsection “2.1. Silver nanoparticles”, it would be helpful to provide a brief explanation of the silver nanoparticles biosynthesis method.
Response: It has been included as suggested.
- Before explaining the MTT assay, the cell culture procedure should be thoroughly explained. The MTT assay should be explained as well.
Response: It has been included as suggested.
- The number of replicates for each test should be specified.
Response: It has been corrected as suggested.
- Reference 51 is not available because the article is “in press.” Therefore, the quality of the produced silver nanoparticles cannot be evaluated based on the current manuscript. It is strongly advised that some of the results concerning the characterization of the prepared nanoparticles be included.
Response: It has been included as suggested.
- Figure 3 is of poor quality.
Response: It has been corrected as suggested.
- Statistically significant changes should be indicated in figure 2.
Response: It has been corrected as suggested.
- The control and experimental groups of the MTT assay and scratch assay should be defined in the “Materials and Methods” section.
Response: It has been corrected as suggested.
- It would be interesting if the authors discussed the limitations of using silver nanoparticles and their dose-dependent toxicity in the “Discussion” section. (For the reference: https://bmcnephrol.biomedcentral.com/articles/10.1186/s12882-021-02428-5)
Response: It has been corrected/included as suggested.
- Most of the references are not up-to-date. The authors should use the most recently published articles to provide background on the studied subjects and to discuss the findings.
Response: It has been corrected as suggested.
- “et al.” should be written in italics throughout the manuscript.
Response: It has been corrected as suggested.

Round 2
Reviewer 1 Report
The manuscript is improved and the changes are incorporated. Needs final check for spellings before acceptance.
Reviewer 2 Report
The corrections and additions introduced by the authors improved the structure and quality of the manuscript. I have no further suggestions.